# Thresholds for clinically important deterioration versus improvement in COPD health status: results from a randomised controlled trial in pulmonary rehabilitation and an observational study during routine clinical practice

Harma Johanna Alma,[1,2] Corina de Jong,[1,2] Danijel Jelusic,[3] Michael Wittmann,[3] Michael Schuler,[4] Robbert Sanderman,[5,6] Konrad Schultz,[3] Janwillem Kocks,[1,2] Thys van der Molen[1,2]

For numbered affiliations see end of article.

**Correspondence to**
Mrs Harma Johanna Alma;
h.j.alma@umcg.nl

## ABSTRACT

**Objectives** Chronic Obstructive Pulmonary Disease (COPD) is a progressive disease. Preventing deterioration of health status is therefore an important therapy goal. (Minimal) Clinically Important Differences ((M)CIDs) are used to interpret changes observed. It remains unclear whether (M)CIDs are similar for both deterioration and improvement in health status. This study investigates and compares these clinical thresholds for three widely-used questionnaires.

**Design and setting** Data were retrospectively analysed from an inhouse 3-week pulmonary rehabilitation (PR) randomised controlled trial in the German Klinik Bad Reichenhall (study 1), and observational research in Dutch primary and secondary routine clinical practice (RCP) (study 2).

**Participants** Patients with COPD aged ≥18 years (study 1) and aged ≥40 years (study 2) without respiratory comorbidities were included for analysis.

**Primary outcomes** The COPD Assessment Test (CAT), Clinical COPD Questionnaire (CCQ) and St George's Respiratory Questionnaire (SGRQ) were completed at baseline and at 3, 6 and 12 months. A Global Rating of Change scale was added at follow-up. Anchor-based and distribution-based methods were used to determine clinically relevant thresholds.

**Results** In total, 451 patients were included from PR and 207 from RCP. MCIDs for deterioration ranged from 1.30 to 4.21 (CAT), from 0.19 to 0.66 (CCQ), and from 2.75 to 7.53 (SGRQ). MCIDs for improvement ranged from −3.78 to −1.53 (CAT), from −0.50 to −0.19 (CCQ), and from −9.20 to −2.76 (SGRQ). Thresholds for moderate improvement versus deterioration ranged from −5.02 to −3.29 vs 3.89 to 8.14 (CAT), from −0.90 to −0.72 vs 0.42 to 1.23 (CCQ), and from −15.85 to −13.63 vs 7.46 to 9.30 (SGRQ).

**Conclusions** MCID ranges for improvement and deterioration on the CAT, CCQ and SGRQ were somewhat similar. However, estimates for moderate and large change

## Strengths and limitations of this study

► Our study is the first dedicated investigation of (minimal) clinically important differences ((M)CIDs) for deterioration on chronic obstructive pulmonary disease (COPD) health status tools in comparison with thresholds for improvement.

► Our study used a combination of anchor-based and distribution-based methods to determine clinically relevant thresholds for both deterioration and improvement.

► Our study investigated clinically relevant thresholds in two different study settings—pulmonary rehabilitation (PR) and routine clinical practice (RCP)—by using data from various follow-up periods to minimise the possible impact of the recall period.

► Our study included a limited number of patients with deterioration after PR and during RCP, and a limited number of patients indicating moderate and large change in health status.

► Our study resulted in broad ranges and wide CIs for (M)CIDs of COPD health status tools, requiring possibly larger sample sizes for more accuracy.

varied and were inconsistent. Thresholds differed between study settings.

**Trial registration number** Routine Inspiratory Muscle Training within COPD Rehabilitation trial: #DRKS00004609; MCID study: #UMCG201500447.

## INTRODUCTION

The use of health status questionnaires is recommended by the Global initiative for Chronic Obstructive Lung Disease (GOLD) for the assessment, evaluation

and management of patients with chronic obstructive pulmonary disease (COPD).[1] The COPD Assessment Test (CAT),[2] the Clinical COPD Questionnaire (CCQ)[3] and the St George's Respiratory Questionnaire (SGRQ)[4] are frequently used patient-reported health status tools important for clinical practice and scientific research,[5] especially since the burden of COPD is high worldwide.[6 7]

Various studies have examined clinically relevant thresholds for change on the CAT, CCQ and SGRQ in order to be able to evaluate and interpret treatment effects.[8–18] The minimal clinically important difference (MCID) is a parameter that quantifies this threshold. It has been defined as *'the smallest difference in score, which patients perceive as beneficial and which would mandate a change in the patient's management'*.[19] MCIDs are particularly interesting for health status questionnaires, where a change in its score is not intuitively meaningful. Change exceeding the level of the MCID can be considered clinically relevant, thus justifying therapy and help developing guidelines. It is pivotal that clinically relevant thresholds for change on a health status tool are rigorously studied and analysed carefully.

Most clinical studies that determine the MCID of patient-reported outcomes (PROs) are executed in the context of an intervention such as pharmacotherapy or pulmonary rehabilitation (PR). This usually results in an improvement in the patients' health-related quality of life (HRQoL). MCIDs for improvement have thus been investigated; however, there is a lack of evidence for the MCIDs for deterioration.[20] It remains unclear and debated on to what extent clinically relevant thresholds for improvement should be similar to those for deterioration.[21–24] Certain studies outside the field of COPD have analysed the MCIDs of PROs and found evidence that values for improvement differed from deterioration.[25–29] On the other hand, there is also evidence that thresholds might be similar.[30] Interpreting worsening of HRQoL is of major importance, since one needs to differentiate between real worsening of patients' status and random variations. Furthermore, the effects of therapy may also halt further deterioration especially for a progressive chronic disease like COPD. So no relevant worsening or a reduction in clinically relevant deterioration over time might also be considered a success of therapy and in clinical trials.[31]

In COPD health status, the estimated MCID for the CAT score is 2.00–3.00 units,[11–15 20] for the CCQ score 0.40–0.50 units[8–13 20] and for the SGRQ score 4.00–8.00 units.[12 16–18 20] This is valid for improvement only, as there were too few patients with deterioration to investigate. There are currently no studies that specifically investigate clinically relevant thresholds for deterioration on these PROs. It is however worrying that up to date, multiple studies included the MCIDs of these COPD health status instruments for improvement to interpret deterioration in clinical trials.[32–34] This study therefore aims to determine and compare clinically relevant thresholds for deterioration and improvement on the COPD health status questionnaires CAT, CCQ and SGRQ in both a PR and routine clinical practice (RCP) setting.

## PATIENTS AND METHODS

### Study subjects

This study was a retrospective analysis of data obtained from two prospective clinical trials. Study 1 was a secondary analysis of a subsample from the Routine Inspiratory Muscle Training within COPD Rehabilitation (RIMTCORE) real-life randomised controlled trial in the Klinik Bad Reichenhall, Center for Rehabilitation, Pulmonology and Orthopedics in Germany.[12 35] Patients were recruited on arrival in the clinic between February 2013 and July 2014. Participants were included if they had COPD GOLD II–IV, were aged ≥18 years and gave informed consent.[12 35] Exclusion criteria were the presence of other respiratory comorbidities (eg, bronchiectasis, asthma, history of bronchial carcinoma, sarcoidosis, tuberculosis) or alpha-1-antitrypsin deficiency.

Study 2 (MCID study) was an observational trial of patients with COPD GOLD I–IV aged ≥40 years without other respiratory comorbidities or alpha-1-antitrypsin deficiency. Patients were recruited from Dutch primary and secondary RCP between September 2015 and September 2016. Patients were approached via multiple general practices, hospitals and the Dutch patient lung federation.

### Patient and public involvement

In both studies, patients and the public have not actively been involved during the design of the study nor in the assessment of the burden. Summary results are disseminated to participating patients after completion.

### Study design and data collection

Patients in study 1 participated in an intensive, 3-week, full-day inpatient PR programme tailored to the patient's individual needs. Details have been presented previously.[12 35] Patient descriptives and postbronchodilator spirometry were collected at baseline and discharge in the clinic. Patients in study 2 received routine care from their physician according to national treatment guidelines. Evaluation of health status over a 12-month period was the primary measurement outcome. Patient descriptives and spirometry data were obtained at baseline. Spirometry results were obtained via the including physician after approval of the participant.

The primary outcomes selected from both prospective studies for this retrospective analysis were the CAT (no recall period), CCQ (weekly version) and SGRQ (monthly version). In study 1, these questionnaires were collected at baseline, at PR discharge and during follow-up at 3, 6, 9 and 12 months. Baseline and discharge measurements were taken in the clinic, where patients were blinded to their baseline scores. Follow-up questionnaires were sent by mail. In study 2, all questionnaires were sent by mail and scored at

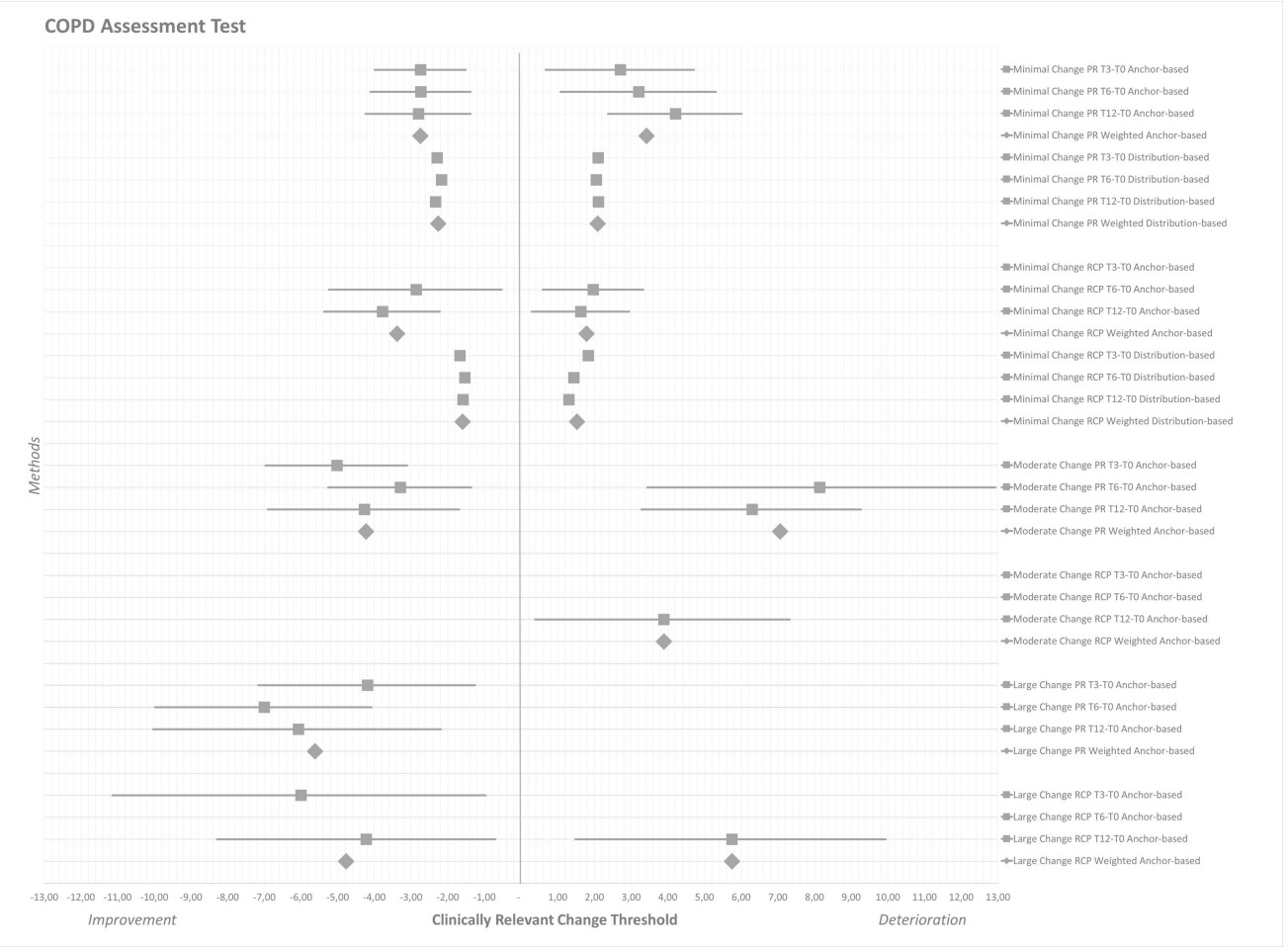

**Figure 1** Forest plot of clinically relevant thresholds for improvement and deterioration on the COPD Assessment Test. Data are presented as mean estimates (squares) including 95% CI (horizontal lines). Estimates from the half SD analysis are represented as single squares. Weighted mean estimates are presented as larger diamonds. Data are separated as minor, moderate and large improvement thresholds (left half), versus minor and moderate deterioration thresholds (right half). COPD, chronic obstructive pulmonary disease; PR, pulmonary rehabilitation; RCP, routine clinical practice. T0, baseline measurement; T3, 3-month follow-up; T6, 6-month follow-up; T12, 12-month follow-up.

home at baseline and at 3, 6 and 12 months. For this retrospective analysis, baseline and follow-up scores at 3, 6 and 12 months were included, to allow for sufficient time for deterioration in HRQoL, to include various time periods of measurement and to allow for comparison between both study settings.

The CAT is an eight-item, one-dimensional scale with item scores ranging 0–5 (0: no impairment; 5: maximum impairment) and a total score summing up to a maximum of 40.[2] The CCQ consists of 10 items scoring 0–6 (0: no impairment; 6: maximum impairment).[3] The items cover the domains symptoms (four items), functional status (four items) and mental status (two items). Total and domain scores on the CCQ derive from adding up relevant item scores and dividing this by the number of items. The SGRQ has 50 items classified into the domains symptoms (8 items), activities (16 items) and impact (26 items).[4] Domain and total SGRQ scores can range from 0 to 100 (0: no impairment; 100: maximum impairment). A 15-point Likert scale anchor question (Global Rating of Change, GRC) was scored

retrospectively by the patient at each follow-up visit in both data sets. The GRC required patients to assess their COPD health status compared with baseline. The answers were marked on a scale from −7 to +7, ranging from *very much worse* to *very much better* and 0 equalling *no change*.[36 37]

### Study methods

All change scores for the total scores of the CAT, CCQ and SGRQ were calculated as the difference between baseline and the respective follow-up visit (3, 6 and 12 months). Negative change on all questionnaires represented improvement, and positive change deterioration. First, in the anchor-based approach, changes on the health status instruments were classified using the corresponding score on the GRC. Scores of 0 and ±1 on the GRC indicated *no change*; scores of ±2 and ±3 represented a *minimal improvement/deterioration*; scores of ±4 and ±5 were summarised as a *moderate improvement/deterioration*; and scores of ±6 and ±7 indicated a *large improvement/deterioration*.[36 37] MCID estimates for

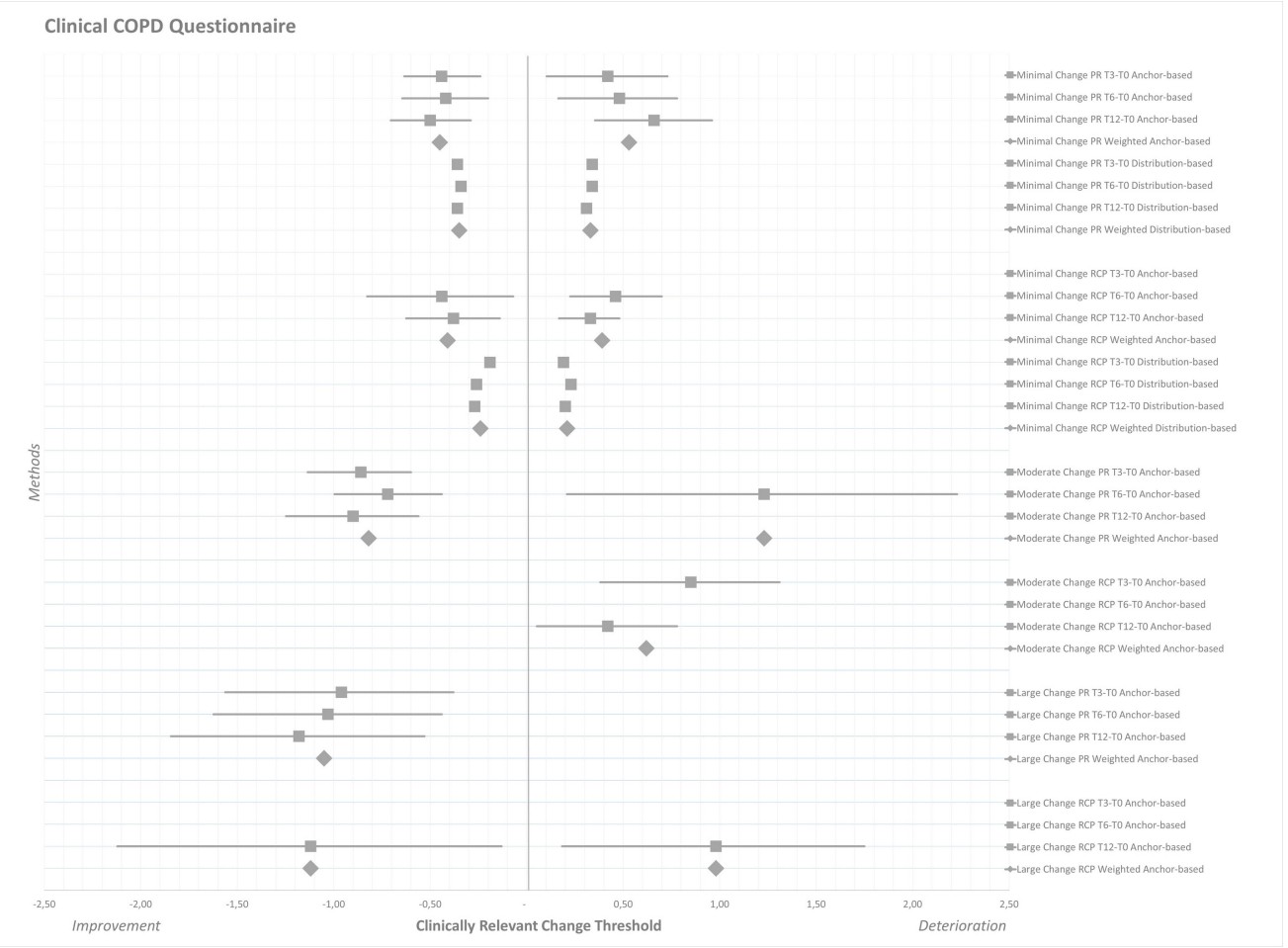

**Figure 2** Forest plot of clinically relevant thresholds for improvement and deterioration on the Clinical COPD Questionnaire. Data are presented as mean estimates (squares) including 95% CI (horizontal lines). Estimates from the half SD analysis are represented as single squares. Weighted mean estimates are presented as larger diamonds. Data are separated as minor, moderate and large improvement thresholds (left half), versus minor and moderate deterioration thresholds (right half). COPD, chronic obstructive pulmonary disease; PR, pulmonary rehabilitation; RCP, routine clinical practice; T0, baseline measurement; T3, 3-month follow-up; T6, 6-month follow-up; T12, 12-month follow-up.

both improvement and deterioration on the CAT, CCQ and SGRQ were calculated as the mean change scores including 95% CI of those patients indicating a minimal improvement/deterioration (±2 and ±3) on the GRC for each follow-up visit, verifying normality of distribution. Mean estimates including 95% CI were determined in a similar way for patients indicating no change (GRC 0 and ±1), moderate change (GRC ±4 and ±5) and large change (GRC ±6 and ±7). Second, the distribution-based method half SD (0.5 SD) of the change score was calculated for improved and deteriorating health status patients at respective follow-up visits.[38]

### Data analysis
Data analysis was performed using SPSS V.24.0. Descriptives were evaluated at baseline for either frequencies with percentages (%), mean with SD or median with range. This was depending on the variable characteristics and/or normality of distribution. Health status data on the CCQ, CAT and SGRQ were evaluated at baseline (T0), 3 months (T3), 6 months (T6) and after

12 months (T12). Normality of distribution was verified using skewness and kurtosis. Values between −1 and +1 were considered indicative for normality. Data were checked for floor and ceiling effects, defined as over 15% of patients scoring in the lowest and highest 10% of the maximum scale range.[39] Mean and SD (or median and range) were calculated at each measurement moment for all patients, as well as specifically for patients with improved and deteriorated health status scores. Baseline scores were compared between improving and deteriorating patients, and tested using independent t-tests after verifying normality of distribution. Baseline scores were compared between both data sets (PR vs RCP) using independent t-tests, Mann-Whitney U tests or $\chi^2$ tests depending on the variable characteristic and/or normality of distribution. Health status change scores were all calculated in comparison with baseline. Follow-up scores were compared with baseline to test for significance of change using paired t-tests verifying normality of distribution.

In order to determine clinically relevant thresholds for change, first correlations between the GRC and the CCQ, CAT and SGRQ were assessed using Pearson or Spearman correlation coefficients depending on normality of distribution. Correlations needed to be ≥0.30 (preferably ≥0.50) to be eligible as anchor.[22] Correlations were assessed between GRC and questionnaire change scores, and between GRC, baseline and follow-up questionnaire score to assess for a possible *response shift*. Next, participants were categorised according to their GRC score at each follow-up. Mean changes (95% CI) for each respective category were determined to define thresholds for clinically relevant change. Significance of change for each GRC class at the respective follow-up visit was compared with baseline and assessed with paired t-tests verifying normality of the data. Last, the 0.5 SD of the change score was determined for patients with improved and deteriorating health status change scores separately at each follow-up. Thresholds were compared between both study settings (PR vs RCP).

An absolute overall weighted mean MCID estimate for both improvement and deterioration was calculated at the end by multiplying the number of observations (n) at each follow-up visit times the MCID estimate for that period. The sum was divided by the total number of observations. Anchor-based and distribution-based approaches had similar weights. Estimates for improvement and deterioration were compared visually in a plot.

## RESULTS
### Patient characteristics
Study 1 included 451 patients with completed baseline data (table 1).[12 35] During follow-up 355 patients (78.7%) had completed data at T3, 319 patients (70.7%) at T6 and 309 patients (68.5%) at T12. During the 12-month follow-up, 8 patients passed away, 41 dropped out at own request, and a varying number of non-response was present. Study 2 included 207 patients with full baseline data (table 1), of whom 201 (97.1%) completed T3, 186 (89.9%) T6 and 177 (85.6%) T12. Four patients died, 12 patients discontinued at own request, and a various number of non-response was present.

There were no significant baseline differences between completers and non-completers of the 12-month follow-up in both studies, except that significantly more women (28.4%) compared with men (10.0%) did not complete the follow-up during RCP. Significant differences in age, forced expiratory volume in 1 s percentage predicted (FEV1%pred) and health status were observed between both studies (table 1).

### Health status scores for improvement and deterioration
In study 1 and study 2, CAT, CCQ and SGRQ total were normally distributed at baseline and follow-up. Completed pairs of change scores (follow-up vs

**Table 1** Baseline patient characteristics

| | Study 1: PR | Study 2: RCP | Significance testing (p value) |
|---|---|---|---|
| n | 451 | 207 | – |
| Age (years)† | 57.87±6.56 | 66.69±7.91 | <0.001* |
| Gender (male)‡ | 293 (65.0) | 121 (58.5) | 0.507 |
| FEV1%pred† | 50.40±15.11 | 57.06±21.96 | 0.001* |
| GOLD I‡ | – | 35 (17.4) | 0.199 |
| GOLD II | 227 (50.3) | 80 (39.8) | |
| GOLD III | 176 (39.0) | 61 (30.3) | |
| GOLD IV | 48 (10.6) | 25 (12.4) | |
| Smoking pack years† | 40 (30–50) | 37.5 (22.50–51.25) | 0.081 |
| CAT total† | 20.23±7.33 | 18.32±7.22 | 0.002* |
| CCQ total† | 2.86±1.17 | 2.12±1.02 | <0.001* |
| CCQ symptoms† | 2.87±1.24 | 2.48±1.03 | <0.001* |
| CCQ functional status† | 2.86±1.34 | 2.28±1.40 | <0.001* |
| CCQ mental status† | 2.86±1.74 | 1 (0–1.50) | <0.001* |
| SGRQ total† | 50.69±17.33 | 42.88±19.16 | <0.001* |
| SGRQ symptoms† | 63.66±21.77 | 48.04±24.16 | <0.001* |
| SGRQ activities† | 63.58±19.82 | 61.48±21.10 | 0.259 |
| SGRQ impact† | 39.21±18.81 | 30.52±19.73 | <0.001* |
| mMRC† | 2 (2–4) | 1 (1–2) | <0.001* |

*Significance testing at p<0.05 using unpaired t-tests, Mann-Whitney U tests or $\chi^2$ tests.
†Data are expressed as mean±SD or median (IQR).
‡Data are expressed as frequencies (% of total).
CAT, COPD Assessment Test; CCQ, Clinical COPD Questionnaire; COPD, chronic obstructive pulmonary disease; FEV1%pred, forced expiratory volume in 1 s % predicted; GOLD, Global Initiative for Chronic Obstructive Lung Disease; mMRC, modified Medical Research Council Dyspnea Scale; n, number of patients; PR, pulmonary rehabilitation; RCP, routine clinical practice; SGRQ, St George's Respiratory Questionnaire.

baseline) were included (pairwise deletion). Floor and ceiling effects were negligible. Mean health status baseline scores were significantly different for PR and RCP (table 1). Overall, 58%–59% of patients had *improved* health status scores (negative change) at T12 after PR, compared with 44%–47% during RCP (table 2). After PR mean changes observed on the CAT questionnaire at T12 were −5.45±4.66 for improvers and 5.47±4.22 for patients who deteriorated; on the CCQ questionnaire −0.87±0.72 for improvement and 0.83±0.62 for deterioration; and on the SGRQ questionnaire −13.83±10.43

**Table 2** Health status baseline and change scores for all, improved and deteriorated patients during PR and RCP

| | Change after 3 months | n | Change after 6 months | n | Change after 12 months | n |
|---|---|---|---|---|---|---|
| **CAT** | | | | | | |
| All patients PR | −1.44* (−2.16 to −0.71) | 354 | −0.91* (−1.66 to −0.16) | 319 | −0.89* (−1.68 to −0.11) | 309 |
| Improvement PR | −5.45±4.57 | 227 (64.1) | −5.49±4.33 | 184 (57.7) | −5.45±4.66 | 180 (58.3) |
| Deterioration PR | 5.75±4.20 | 127 (35.9) | 5.33±4.10 | 135 (42.3) | 5.47±4.22 | 129 (41.7) |
| All patients RCP | 0.30 (−0.42 to +1.02) | 201 | 0.18 (−0.53 to +0.90) | 186 | 0.14 (−0.59 to +0.87) | 177 |
| Improvement RCP | −4.04±3.33 | 102 (50.7) | −4.64±3.05 | 81 (43.5) | −4.53±3.15 | 79 (44.6) |
| Deterioration RCP | 4.23±3.66 | 83 (41.3) | 3.76±2.88 | 91 (48.9) | 3.88±2.59 | 86 (48.6) |
| No change RCP | – | 16 (8.0) | – | 14 (7.5) | – | 12 (6.8) |
| **CCQ total** | | | | | | |
| All patients PR | −0.26* (−0.37 to −0.15) | 355 | −0.11 (−0.23 to +0.01) | 319 | −0.16* (−0.28 to −0.04) | 309 |
| Improvement PR | −0.88±0.71 | 225 (63.4) | −0.84±0.68 | 181 (56.7) | −0.87±0.72 | 180 (58.3) |
| Deterioration PR | 0.82±0.68 | 130 (36.6) | 0.84±0.67 | 138 (43.3) | 0.83±0.62 | 129 (41.7) |
| All patients RCP | 0.00 (−0.09 to +0.08) | 200 | 0.00 (−0.10 to +0.10) | 185 | −0.02 (−0.12 to +0.09) | 174 |
| Improvement RCP | −0.45±0.37 | 96 (48.0) | −0.52±0.51 | 87 (47.0) | −0.54±0.54 | 77 (44.3) |
| Deterioration RCP | 0.50±0.38 | 89 (44.5) | 0.56±0.46 | 80 (43.2) | 0.51±0.39 | 88 (50.6) |
| No change RCP | – | 15 (7.5) | – | 18 (9.7) | – | 9 (5.2) |
| **SGRQ total** | | | | | | |
| All patients PR | −5.35* (−6.92 to −3.78) | 350 | −4.85* (−6.47 to −3.23) | 312 | −3.94* (−5.67 to −2.21) | 306 |
| Improvement PR | −13.11±9.65 | 237 (67.7) | −13.51±9.88 | 193 (61.9) | −13.83±10.43 | 180 (58.8) |
| Deterioration PR | 10.93±10.18 | 113 (32.3) | 8.19±8.92 | 119 (38.1) | 10.19±8.94 | 126 (41.2) |
| All patients RCP | −0.52 (−1.77 to +0.73) | 198 | −1.34 (−2.76 to +0.07) | 184 | −0.87 (−2.60 to +0.86) | 174 |
| Improvement RCP | −6.61±5.58 | 97 (49.0) | −7.91±5.52 | 75 (40.8) | −7.74±9.51 | 81 (46.6) |
| Deterioration RCP | 7.36±5.49 | 101 (51.0) | 7.78±6.18 | 108 (58.7) | 8.46±7.06 | 92 (52.9) |
| No change RCP | – | 0 | – | 1 (0.5) | – | 1 (0.6) |

Change was calculated compared with baseline. Negative change represents improvement for CAT, CCQ and SGRQ. Change scores for all patients reported as mean (95% CI). Change scores for improvement and deterioration are presented as mean±SD.

*Paired t-tests were significant at p<0.05 testing follow-up versus baseline measurements.

CAT, COPD Assessment Test; CCQ, Clinical COPD Questionnaire; COPD, chronic obstructive pulmonary disease; n, number of patients; PR, pulmonary rehabilitation; RCP, routine clinical practice; SGRQ, St George's Respiratory Questionnaire.

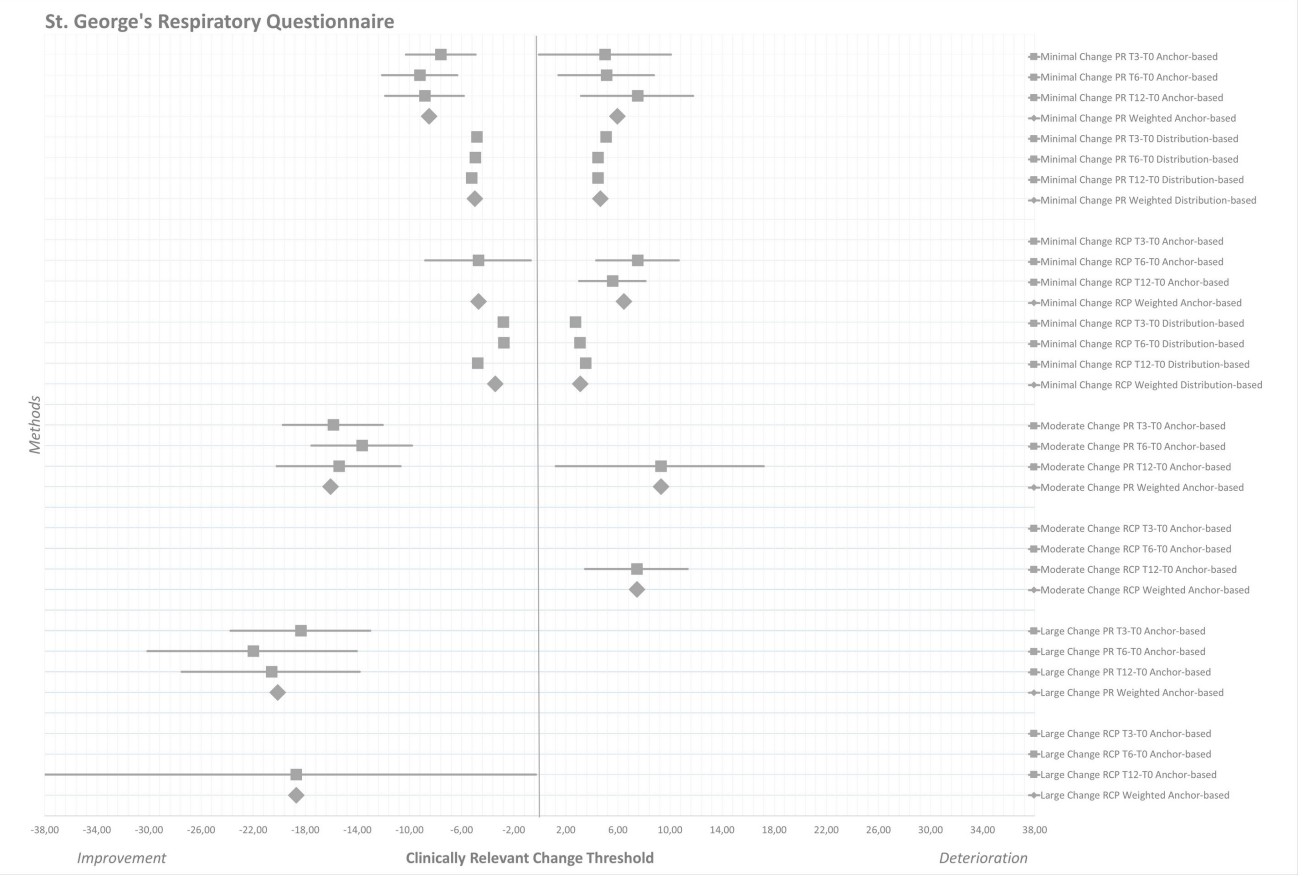

**Figure 3** Forest plot of clinically relevant thresholds for improvement and deterioration on the St George's Respiratory Questionnaire. Data are presented as mean estimates (squares) including 95% CI (horizontal lines). Estimates from the half SD analysis are represented as single squares. Weighted mean estimates are presented as larger diamonds. Data are separated as minor, moderate and large improvement thresholds (left half), versus minor and moderate deterioration thresholds (right half). PR, pulmonary rehabilitation; RCP, routine clinical practice; T0, baseline measurement; T3, 3-month follow-up; T6, 6-month follow-up; T12, 12-month follow-up.

for improvers and 10.19±8.94 for patients who deteriorated (table 2). For RCP, these estimates were for the CAT −4.53±3.15 for improvement and 3.88±2.59 for deterioration; for the CCQ −0.54±0.54 for improvement and 0.51±0.39 for deterioration; and for the SGRQ −7.74±9.51 for improvement and 8.46±7.06 for deterioration (table 2).

There were no baseline differences in terms of age, gender and GOLD classification between patients with improved health status and those who deteriorated at T12 in both studies. Patients with a worse (read higher) CAT, CCQ or SGRQ baseline score prior to PR had significantly more improved health status after 1 year. Patients who improved during RCP had a significantly better baseline FEV1%pred.

### Clinically important improvement versus deterioration

Significant correlations between the health status change scores and the GRC ranged, respectively, for study 1 from −0.33 to −0.41 (CAT), from −0.42 to −0.47 (CCQ), and from −0.48 to −0.54 (SGRQ) (table 3). These ranges were for study 2, respectively, from −0.29 to −0.37, from −0.38 to −0.48, and from −0.35 to −0.44. GRC scores had stronger correlations

with the respective follow-up health status score compared with baseline and change scores for both studies.

Tables 4–6 and figures 1–3 present the clinically relevant thresholds for minimal, moderate and large changes on the CAT, CCQ and SGRQ during PR and RCP. On the CAT anchor-based and distribution-based estimates ranged from −2.80 to −2.17 (weighted mean −2.51) for minimal improvement and from 2.05 to 4.21 for minimal deterioration (weighted mean 2.76) during PR (table 4, figure 1). These ranges were, respectively, from −3.78 to −1.53 (weighted mean −2.49) and from 1.30 to 1.97 (weighted mean 1.65) during RCP. On the CCQ minimal clinically important improvements were determined at −0.50 to −0.34 (weighted mean −0.40) for PR and at −0.44 to −0.19 (weighted mean −0.33) for RCP (table 5, figure 2). These thresholds for deterioration were from 0.31 to 0.66 (weighted mean 0.43) during PR and from 0.19 to 0.46 (weighted mean 0.30) during RCP. On the SGRQ estimates ranged from −9.20 to −4.83 (weighted mean −6.74) for minimal improvement and from 4.46 to 7.52 for minimal deterioration (weighted mean 5.31) during PR (table 6, figure 3). These ranges were, respectively, from −4.76 to −2.76

**Table 3** Correlations between health status (change) scores and the GRC

| | GRC T3–T0 | | GRC T6–T0 | | GRC T12–T0 | |
| --- | --- | --- | --- | --- | --- | --- |
| | PR (n=355) | RCP (n=201) | PR (n=319) | RCP (n=186) | PR (n=309) | RCP (n=177) |
| CAT change score | −0.33* | −0.29* | −0.40* | −0.30* | −0.41* | −0.37* |
| CAT T0 | −0.31* | −0.11 | −0.25* | −0.22* | −0.34* | −0.22* |
| CAT T3 | **−0.56*** | −0.31* | **−0.50*** | −0.31* | **−0.50*** | −0.33* |
| CAT T6 | – | – | **−0.55*** | −0.40* | **−0.59*** | −0.34* |
| CAT T12 | – | – | – | – | **−0.64*** | −0.48* |
| CCQ change score | −0.42* | −0.38* | −0.44* | −0.40* | −0.47* | −0.48* |
| CCQ T0 | −0.26* | −0.14* | −0.19* | −0.22* | −0.29* | −0.23* |
| CCQ T3 | **−0.61*** | −0.35* | **−0.52*** | −0.26* | **−0.54*** | −0.33* |
| CCQ T6 | – | – | **−0.56*** | −0.43* | **−0.59*** | −0.39* |
| CCQ T12 | – | – | – | – | **−0.66*** | **−0.51*** |
| SGRQ change score | −0.48* | −0.35* | **−0.51*** | −0.33* | **−0.54*** | −0.44* |
| SGRQ T0 | −0.28* | −0.13 | −0.24* | −0.20* | −0.32* | −0.22* |
| SGRQ T3 | **−0.62*** | −0.29* | **−0.56*** | −0.25* | **−0.58*** | −0.28* |
| SGRQ T6 | – | – | **−0.61*** | −0.35* | **−0.62*** | −0.35* |
| SGRQ T12 | – | – | – | – | **−0.69*** | **−0.51*** |

Data reported as Pearson or Spearman correlation coefficients between the health status (change) scores and the GRC anchor question.
Correlations ≥0.50 are highlighted bold.
*Correlations are significant at p<0.05.
CAT, COPD Assessment Test; CCQ, Clinical COPD Questionnaire; COPD, chronic obstructive pulmonary disease; GRC, Global Rating of Change; n, number of patients; PR, pulmonary rehabilitation; RCP, routine clinical practice; SGRQ, St George's Respiratory Questionnaire; T0, baseline measurement; T3, 3-month follow-up; T6, 6-month follow-up; T12, 12-month follow-up.

(weighted mean −4.06) and from 2.75 to 7.53 (weighted mean 4.78) during RCP.

## DISCUSSION

### Summary of main findings

Using both anchor-based and distribution-based methods, the *weighted MCIDs* for improvement and deterioration on the CAT were, respectively, −2.51 vs 2.76 during PR, and −2.49 vs 1.65 during RCP. These thresholds for improvement and deterioration on the CCQ were, respectively, −0.40 vs 0.43 during PR and −0.33 vs 0.30 during RCP. MCIDs for the SGRQ were, respectively, −6.74 vs 5.31 during PR and −4.06 vs 4.78 during RCP for improvement and deterioration. Estimates for minimal clinically important improvement and deterioration were overall somewhat similar; however, absolute MCIDs differed between PR and RCP. Thresholds for *moderate* and *large* improvement and deterioration differed from each other, as well as between study settings.

### Interpretation of findings

Little evidence exists whether MCIDs for improvement are similar for deterioration.[21 23 40] Jaeschke *et al*[19] were the first to determine the MCID of a health status tool using a 15-point GRC combining both improved and deteriorated patients with COPD into one group of minimally changed participants. Juniper *et al*[37] elaborated on this by separating minimally improved patients from deterioration in asthma, but only a limited number of patients indicated deterioration and no conclusions on the MCID of deterioration were drawn. Outside the field of COPD, Crosby *et al* and de Vet *et al*[21 40] stated that some studies demonstrated that a smaller MCID for improvement was required compared with deterioration. The current study does not confirm this, although MCIDs seemed smaller for RCP patients compared with PR. Patients experienced more change (hence larger absolute MCIDs) during intervention, possibly as a result of treatment. In RCP, smaller changes may be noted and regarded as relevant for the patient. Up to now it remains unclear, whether the reported differences between PR and RCP are a rehab-specific finding or generally as a result of intervention. Overall, the absolute values for the MCIDs for improvement versus deterioration did not seem to differ much here, with the exception of the SGRQ during PR.

The ranges found in this study for the MCID of the CAT (*improvement −3.78 to −1.53; deterioration 1.30 to 4.21*) matched with estimates found in other studies.[11–15 20] Two

**Table 4** Estimates for clinically relevant thresholds for improvement and deterioration on the CAT

| CAT | T3–T0 | | T6–T0 | | T12–T0 | | Weighted threshold | |
|---|---|---|---|---|---|---|---|---|
| Change | Improvement | Deterioration | Improvement | Deterioration | Improvement | Deterioration | Improvement | Deterioration |
| **Minimal change** | | | | | | | | |
| Anchor-based PR (n) | 107 | 36 | 96 | 42 | 88 | 43 | 291 | 121 |
| Anchor-based PR | –2.74 | 2.71 | –2.73 | 3.21 | –2.80 | 4.21 | –2.75 | 3.42 |
| Anchor-based RCP (n) | 12 | 27 | 14 | 36 | 18 | 46 | 32 | 82 |
| Anchor-based RCP | – | – | –2.86 | 1.97 | –3.78 | 1.63 | –3.38 | 1.78 |
| Distribution-based PR (n) | 227 | 127 | 184 | 135 | 180 | 129 | 591 | 391 |
| Distribution-based PR | –2.29 | 2.10 | –2.17 | 2.05 | –2.33 | 2.11 | –2.26 | 2.09 |
| Distribution-based RCP (n) | 102 | 83 | 81 | 91 | 79 | 86 | 262 | 260 |
| Distribution-based RCP | –1.67 | 1.83 | –1.53 | 1.44 | –1.58 | 1.30 | –1.60 | 1.52 |
| **Moderate change** | | | | | | | | |
| Anchor-based PR (n) | 51 | 9 | 45 | 7 | 37 | 10 | 133 | 17 |
| Anchor-based PR | –5.02 | – | –3.29 | 8.14 | –4.27 | 6.30 | –4.23 | 7.06 |
| Anchor-based RCP (n) | 5 | 8 | 12 | 9 | 5 | 9 | – | 9 |
| Anchor-based RCP | – | – | – | – | – | 3.89 | – | 3.89 |
| **Large change** | | | | | | | | |
| Anchor-based PR (n) | 16 | 3 | 12 | 2 | 14 | 3 | 42 | – |
| Anchor-based PR | –4.19 | – | –7.00 | – | –6.07 | – | –5.62 | – |
| Anchor-based RCP (n) | 4 | 3 | 0 | 2 | 9 | 4 | 13 | 4 |
| Anchor-based RCP | –6.00 | – | – | – | –4.22 | 5.75 | –4.77 | 5.75 |

**Table 4** Continued

| CAT Change | T3–T0 Improvement | T3–T0 Deterioration | T6–T0 Improvement | T6–T0 Deterioration | T12–T0 Improvement | T12–T0 Deterioration | Weighted threshold Improvement | Weighted threshold Deterioration |
|---|---|---|---|---|---|---|---|---|
| No change | | | | | | | | |
| Anchor-based PR (n) | 133 | | 115 | | 114 | | 362 | |
| Anchor-based PR | 0.03 | | –0.01 | | –0.33 | | –0.10 | |
| Anchor-based RCP (n) | 141 | | 113 | | 83 | | 337 | |
| Anchor-based RCP | –0.16 | | –0.54 | | –0.47 | | –0.36 | |

Data reported as clinically relevant threshold or n. Negative change represents improvement for all health status instruments. Paired t-tests were applied with significance level at $p<0.05$. Non-significant results were excluded, except for the 'No change' group.
CAT, COPD Assessment Test; n, number of patients; PR, pulmonary rehabilitation; RCP, routine clinical practice; T0, baseline measurement; T3, 3-month follow-up; T6, 6-month follow-up; T12, 12-month follow-up.

studies used a patient-assessed GRC to estimate the MCID of the CAT.[14 15] However, no results were reported for worsened patients or the numbers of patients were too few. Other anchor-based methods suggested that a change of one point on the CAT might represent the MCID for deterioration.[14] The weighted thresholds for minimal clinically relevant improvement *(–2.51 in PR and –2.49 in RCP)* seemed somewhat comparable with the ones for deterioration *(2.76 in PR and 1.65 in RCP)* in the current study, except for deterioration during RCP. As CAT allows only integer scores,[2] a change of three points seems a valid threshold for improvement and deterioration, although the MCID for deterioration in RCP could be closer to two points. Thresholds for moderate improvement *(–4.23 in PR)* and deterioration *(7.06 in PR and 3.89 in RCP)* turned out less similar. The number of patients moderately deteriorating was low and differences were observed between both study settings. Moderate change might be experienced with a change on the CAT score of four to seven points. Two previous studies suggested that a cut-off point of four points was identified for acute HRQoL deterioration in clinical practice.[41 42] This would match our estimates for moderate change. The number of patients with a large change was too low with wide CIs to enable valid conclusions.

Regarding the CCQ, the MCID ranges found for both improvement *(–0.50 to –0.19)* and deterioration *(0.19–0.66)* overlapped each other in absolute sense, indicating that estimates for improvement and deterioration may be similar. However, differences were noted between PR *(±0.40)* and RCP *(±0.30)* for both minimal improvement and deterioration. These estimates for the MCID matched with earlier evidence.[8–13] One other study used a GRC to determine the MCID of the CCQ.[8] Unfortunately, no data were available on worsening patients. Thresholds for moderate change on the CCQ were broad *(±0.62 to ±1.23)*. Few patients experienced large changes, but estimates for both types of MCID from both study settings were approximately one point.

Minimal thresholds for improvement *(–9.20 to –2.76)* and deterioration *(2.75–7.53)* on the SGRQ overlapped each other, although more variation was present here. A change of approximately four to seven points for both improvement and deterioration seemed to be the minimal clinically important threshold in the current study. The MCID for improvement during PR *(–6.74)* was larger than for deterioration *(5.31)*; however, CIs for deterioration were wide. Estimates for the thresholds during RCP (4–5 points) were smaller compared with PR (5–7 points). Moreover, the distribution-based estimates turned out smaller than the anchor-based estimates, lowering the absolute weighted MCIDs. Thresholds for moderate improvement and deterioration in the current study were not very similar, ranging absolutely from 7.46 to 16.06 points. Estimates for clinically relevant large HRQoL improvement on the SGRQ ranged from –20 to –18 points for PR and RC, but too few patients were included to draw valid conclusions.

The SGRQ MCID matched to some extent with previous results.[12 16–18 20] Jones *et al*[16 18] published a threshold of four points, which is generally accepted and applied in

**Table 5** Estimates for clinically relevant thresholds for improvement and deterioration on the CCQ

| CCQ | T3–T0 | | T6–T0 | | T12–T0 | | Weighted threshold | |
|---|---|---|---|---|---|---|---|---|
| Change | Improvement | Deterioration | Improvement | Deterioration | Improvement | Deterioration | Improvement | Deterioration |
| Minimal change | | | | | | | | |
| Anchor-based PR (n) | 107 | 36 | 96 | 42 | 88 | 43 | 291 | 121 |
| Anchor-based PR | −0.44 | 0.42 | −0.42 | 0.48 | −0.50 | 0.66 | −0.45 | 0.53 |
| Anchor-based RCP (n) | 12 | 27 | 14 | 36 | 18 | 46 | 32 | 82 |
| Anchor-based RCP | – | – | −0.44 | 0.46 | −0.38 | 0.33 | −0.41 | 0.39 |
| Distribution-based PR (n) | 225 | 130 | 181 | 138 | 180 | 129 | 586 | 397 |
| Distribution-based PR | −0.36 | 0.34 | −0.34 | 0.34 | −0.36 | 0.31 | −0.35 | 0.33 |
| Distribution-based RCP (n) | 96 | 89 | 87 | 80 | 77 | 88 | 260 | 257 |
| Distribution-based RCP | −0.19 | 0.19 | −0.26 | 0.23 | −0.27 | 0.20 | −0.24 | 0.21 |
| Moderate change | | | | | | | | |
| Anchor-based PR (n) | 51 | 9 | 45 | 7 | 37 | 10 | 133 | 7 |
| Anchor-based PR | −0.86 | – | −0.72 | 1.23 | −0.90 | – | −0.82 | 1.23 |
| Anchor-based RCP (n) | 5 | 8 | 12 | 9 | 5 | 9 | – | 17 |
| Anchor-based RCP | – | 0.85 | – | – | – | 0.42 | – | 0.62 |
| Large change | | | | | | | | |
| Anchor-based PR (n) | 16 | 3 | 12 | 2 | 14 | 3 | 42 | – |
| Anchor-based PR | −0.96 | – | −1.03 | – | −1.18 | – | −1.05 | – |
| Anchor-based RCP (n) | 4 | 3 | 0 | 2 | 9 | 4 | 9 | 4 |
| Anchor-based RCP | – | – | – | – | −1.12 | 0.98 | −1.12 | 0.98 |

Continued

**Table 5** Continued

| CCQ | T3–T0 | | T6–T0 | | T12–T0 | | Weighted threshold | |
|---|---|---|---|---|---|---|---|---|
| Change | Improvement | Deterioration | Improvement | Deterioration | Improvement | Deterioration | Improvement | Deterioration |
| No change | | | | | | | | |
| Anchor-based PR (n) | 133 | | 115 | | 114 | | 362 | |
| Anchor-based PR | –0.07 | | 0.17 | | 0.10 | | 0.06 | |
| Anchor-based RCP (n) | 141 | | 113 | | 83 | | 337 | |
| Anchor-based RCP | –0.03 | | –0.10 | | –0.04 | | –0.06 | |

Data reported as clinically relevant threshold or n. Negative change represents improvement for all health status instruments. Paired t-tests were applied with significance level at p<0.05. Non-significant results were excluded, except for the 'No change' group.

CCQ, Clinical COPD Questionnaire; n, number of patients; PR, pulmonary rehabilitation; RCP, routine clinical practice; T0, baseline measurement; T3, 3-month follow-up; T6, 6-month follow-up; T12, 12-month follow-up.

clinical practice. Interestingly, most results in our current study suggest a larger MCID, although estimates from RCP included this four-point estimate. The estimate by Jones *et al* was based on a study using patient preference-based techniques in COPD by applying a five-point patients' judgement of treatment efficacy (salmeterol). This MCID of four points was valid for the group of patients that experienced effective treatment. In addition, a clinicians' five-point GRC was scored, resulting in an MCID of four points. Clinicians' and patients' ratings are however not necessarily similar.[43]

### Strengths and limitations of the current study

This retrospective analysis of two prospective studies was the first to investigate clinically relevant thresholds for minimal, moderate and large changes in COPD health status comparing both improvement and deterioration using a triangulation of both anchor-based and distribution-based methods. There were sufficient correlations between the GRC and respective health status questionnaires as required,[22] although they were still only weak to moderate. It should be noted that correlations were stronger with the follow-up score compared with the baseline and/or change score, possibly due to a response shift. Another strength is that multiple follow-up visits were included to limit possible influence of the period of measurements on the MCID and recall bias.[21 24] Moreover, this study investigated clinically relevant thresholds for both PR and RCP, improving its clinical application and external validity.

Although this is the first study to investigate thresholds for clinically relevant deterioration, still a limited number of patients indicated deterioration in HRQoL after PR and during RCP. This is a major limitation lowering the statistical power of the analysis, especially since sample size calculations were not based on the separate GRC categories. A second limitation is that the found thresholds demonstrate broad ranges with wide CIs, limiting its accuracy and requiring a larger sample size than our current studies had. Third, it should be taken into account that anchor-based and distribution-based approaches each has its own relevance, either based on clinical retrospective assessments or statistical parameters. It is recommended to combine both methods in measuring an instrument's MCID[22]; however, estimates were rather different between these methods.

### Implications for future research and clinical practice

Patients with COPD tend to have worsening HRQoL over time; hence, MCIDs for deterioration have an important implication for clinical practice.[44 45] Clinicians and researchers should be able to judge whether groups of patients were really worsening over time or that change observed was subject to random fluctuation. Preventing clinically relevant deterioration in HRQoL by means of therapy is thus an important goal too. Ideally, more research is needed to validate our thresholds for clinically relevant deterioration on the CAT, CCQ and SGRQ, for instance in studies of other kinds of interventions than PR. One cannot directly transform the thresholds for improvement into those for deterioration. Evidence outside the field of COPD

**Table 6** Estimates for clinically relevant thresholds for improvement and deterioration on the SGRQ

| SGRQ | T3–T0 | | T6–T0 | | T12–T0 | | Weighted threshold | |
|---|---|---|---|---|---|---|---|---|
| Change | Improvement | Deterioration | Improvement | Deterioration | Improvement | Deterioration | Improvement | Deterioration |
| Minimal change | | | | | | | | |
| Anchor-based PR (n) | 107 | 36 | 96 | 42 | 88 | 43 | 291 | 121 |
| Anchor-based PR | −7.58 | 5.01 | −9.20 | 5.14 | −8.82 | 7.52 | −8.49 | 5.95 |
| Anchor-based RCP (n) | 12 | 27 | 14 | 36 | 18 | 46 | 14 | 82 |
| Anchor-based RCP | – | – | −4.70 | 7.53 | – | 5.60 | −4.70 | 6.45 |
| Distribution-based PR (n) | 237 | 113 | 193 | 119 | 180 | 126 | 610 | 358 |
| Distribution-based PR | −4.83 | 5.09 | −4.94 | 4.46 | −5.22 | 4.47 | −4.98 | 4.66 |
| Distribution-based RCP (n) | 97 | 101 | 75 | 108 | 81 | 92 | 253 | 301 |
| Distribution-based RCP | −2.79 | 2.75 | −2.76 | 3.09 | −4.76 | 3.53 | −3.41 | 3.11 |
| Moderate change | | | | | | | | |
| Anchor-based PR (n) | 51 | 9 | 45 | 7 | 37 | 10 | 124 | 10 |
| Anchor-based PR | −15.85 | – | −13.63 | – | −15.40 | 9.30 | −16.06 | 9.30 |
| Anchor-based RCP (n) | 5 | 8 | 12 | 9 | 5 | 9 | – | 9 |
| Anchor-based RCP | – | – | – | – | – | 7.46 | – | 7.46 |
| Large change | | | | | | | | |
| Anchor-based PR (n) | 16 | 3 | 12 | 2 | 14 | 3 | 42 | – |
| Anchor-based PR | −18.33 | – | −21.99 | – | −20.58 | – | −20.13 | – |
| Anchor-based RCP (n) | 4 | 3 | 0 | 2 | 9 | 4 | 9 | – |
| Anchor-based RCP | – | – | – | – | −18.70 | – | −18.70 | – |

Continued

**Table 6** Continued

| SGRQ | T3–T0 | | T6–T0 | | T12–T0 | | Weighted threshold | |
|---|---|---|---|---|---|---|---|---|
| Change | Improvement | Deterioration | Improvement | Deterioration | Improvement | Deterioration | Improvement | Deterioration |
| No change | | | | | | | | |
| Anchor-based PR (n) | 133 | | 115 | | 114 | | 362 | |
| Anchor-based PR | −1.50 | | −0.99 | | −0.06 | | −0.88 | |
| Anchor-based RCP (n) | 141 | | 113 | | 83 | | 337 | |
| Anchor-based RCP | 0.51 | | 0.19 | | 0.10 | | 0.30 | |

Data reported as clinically relevant threshold or n. Negative change represents improvement for all health status instruments. Paired t-tests were applied with significance level at p<0.05. Non-significant results were excluded, except for the 'No change' group.

n, number of patients; PR, pulmonary rehabilitation; RCP, routine clinical practice; SGRQ, St George's Respiratory Questionnaire; T0, baseline measurement; T3, 3-month follow-up; T6, 6-month follow-up; T12, 12-month follow-up.

has found differences. However, in the current study, the estimates turned out rather similar with differing MCIDs between studies. Setting could thus potentially impact the MCID, implying that the results in the current study do not necessarily need to be valid in other settings too.

## CONCLUSIONS

Determining deterioration in HRQoL is of importance, since one needs to differentiate between real worsening of patients' status and random variations. In this study, estimates for clinically relevant thresholds for improvement and deterioration were somewhat similar, but differed between PR and RCP. We would recommend using cut-points of CAT ≥3 (intervention), CAT ≥2 (RCP), CCQ ≥0.40 (intervention), CCQ ≥0.30 (RCP), SGRQ ≥6 (intervention) and SGRQ ≥5 (RCP) for both *minimal* improvement and deterioration. Thresholds for, respectively, *moderate* and *large* changes should be further explored, but could approximately be in the range of, respectively, 4–5 and 5–6 for CAT, 0.80 and 1.00 for CCQ, and 10–15 points and 15–20 points for SGRQ.

**Author affiliations**
[1]Department of General Practice and Elderly Care Medicine, University of Groningen, University Medical Center Groningen, Groningen, The Netherlands
[2]Groningen Research Institute for Asthma and COPD (GRIAC), University of Groningen, University Medical Center Groningen, Groningen, The Netherlands
[3]Center for Rehabilitation, Pulmonology and Orthopedics, Klinik Bad Reichenhall, Bad Reichenhall, Germany
[4]Institute for Clinical Epidemiology and Biometry (ICE-B), Julius-Maximilians-Universität Würzburg, Würzburg, Bayern, Germany
[5]Department of Health Psychology, University of Groningen, University Medical Center Groningen, Groningen, The Netherlands
[6]Department of Psychology, Health and Technology, University of Twente, Enschede, The Netherlands

**Acknowledgements** We are grateful to the Junior Scientific Masterclass of the University of Groningen, who financially supported the research position of the first author. We would also like to acknowledge all participating patients in both the RIMTCORE trial and the MCID study.

**Contributors** KS, MW, DJ and MS planned the RIMTCORE study design and were responsible for data collection. HJA, CdJ, RS, JK and TvdM designed the Dutch observational study on COPD health status in routine clinical practice, as well as the current retrospective analysis of both prospective studies. HJA and CdJ performed the statistical analysis. HJA wrote the first draft, while CdJ, RS, JK and TvdM actively participated in the review process. RS and TvdM supervised and participated in the different steps of the study, as well as in writing. All authors edited the manuscript, participated in and approved the final version of the manuscript.

**Funding** The RIMTCORE trial, including patients in pulmonary rehabilitation, was funded by the Deutsche Rentenversicherung. The Dutch observational study on COPD health status in routine clinical practice, as well as the current combined retrospective analysis of both prospective studies, received financial support from the Junior Scientific Masterclass as part of the University of Groningen.

**Competing interests** HJA, CdJ, DJ, MW, MS and RS have nothing to disclose. JK reports personal fees from Novartis; research grants and personal fees from Boehringer Ingelheim; research grants and personal fees from GlaxoSmithKline (GSK); research grants from Stichting Zorgdraad; personal fees from the International Primary Care Respiratory Group (IPCRG); personal fees from Springer Media; and travel arrangements from Chiesi, GSK and IPCRG, all outside the submitted work. KS received lecture fees from Boehringer, AstraZeneca, Berlin-Chemie, Novartis, Chiesi, Mundipharma, Takeda, GSK and MSD, all outside the submitted work. TvdM reports personal reimbursements from GSK, TEVA, AstraZeneca and Boehringer Ingelheim, and study grants from AstraZeneca and

GSK. After this study was terminated, he became an employee of GSK. None of these stated conflicts of interest are linked to the current manuscript. TvdM developed the Clinical COPD Questionnaire (CCQ) and holds the copyright.

**Patient consent for publication** Not required.

**Ethics approval** All patients in both studies signed informed consent upon participation. The RIMTCORE trial has been approved by the Ethik-Kommission der Bayerischen Landesärztekammer (#12107) and registered in the German Clinical Trials Register (#DRKS00004609). The MCID study has been registered at the University Medical Center Groningen (UMCG) Research Register and evaluated by its Medical Ethical Committee (#201500447).

**Provenance and peer review** Not commissioned; externally peer reviewed.

**Data sharing statement** The data that support the findings of this study are not publicly available. Participating patients in the RIMTCORE trial have only agreed on the availability of their data to the Klinik Bad Reichenhall, their scientific partners in the data analysis and the Committee of the Bavarian State Chamber of Labor in Munich. Participating patients in the MCID study only agreed on the availability of their data to the University Medical Center Groningen (UMCG) and their scientific partners in the data analysis.

**Author note** This study is a secondary retrospective analysis of a subsample from the Routine Inspiratory Muscle Training within COPD Rehabilitation (RIMTCORE) real-life randomised controlled trial (#DRKS00004609) in the Klinik Bad Reichenhall, Center for Rehabilitation, Pulmonology and Orthopedics in Germany; and a primary analysis of all patients participating in the Dutch observational trial (MCID study) on COPD health status in routine clinical practice (UMCG trial #201500447).

**Open access** This is an open access article distributed in accordance with the Creative Commons Attribution 4.0 Unported (CC BY 4.0) license, which permits others to copy, redistribute, remix, transform and build upon this work for any purpose, provided the original work is properly cited, a link to the licence is given, and indication of whether changes were made. See: https://creativecommons.org/licenses/by/4.0/.

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
