## [Reviewer comments · BMJ Open]

ARTICLE DETAILS

TITLE (PROVISIONAL)	Thresholds for Clinically Important Deterioration versus Improvement in COPD Health Status: Results from a Randomized Controlled Trial in Pulmonary Rehabilitation and an Observational Study during Routine Clinical Practice
AUTHORS	Alma, Harma; de Jong, Corina; Jelusic, Danijel; Wittmann, Michael; Schuler, Michael; Sanderman, Robbert; Schultz, Konrad; Kocks, Janwillem; van der Molen, Thys

VERSION 1 - REVIEW

REVIEWER	Dr Anabel Sciriha Physiotherapy Department Faculty of Health Sciences University of Malta
REVIEW RETURNED	29-Aug-2018

GENERAL COMMENTS	Line 7-8 In the Abstract, the introduction to the study requires some slight changes to reflect the reason behind the study and introductory sentence was too abrupt. Line 15-26: Methodology needs to be clearer in the way it reads reflecting a better explanation of how the patients were recruited. Literature Review: In this section, one which looked into the importance of noting deterioration as well as improvements, there is a lack of stating the importance of such a study to the clinical field Lines 37 - 44 There is the need to explain the recruitment of patients better arranging the sentences to read in a better manner. In Line 51, 52 and 54, it is stated that spirometry tests were carried out. How was this done when it is stated a bit before that patients were not actively involved. Discussion and conclusions were well discussed
---

REVIEWER	Rachael Evans University of Leicester, Leicester, UK.
REVIEW RETURNED	14-Nov-2018

GENERAL COMMENTS	Thresholds for Clinically Important Deterioration versus Improvement in COPD health status during Pulmonary Rehabilitation and Routine Clinical Practice: A Retrospective Analysis. The authors present data from two separate studies regarding the MCID of commonly used questionnaires to assess health status in patients with COPD over multiple time points up to 1 year after pulmonary rehabilitation and during routine clinical practice. The paper is overall easy to read, includes a reasonably large data set >650 patients over two different settings and the benefits over existing research are the inclusion of both improvement and deterioration assessed over multiple time points. Both anchor and distribution methods are used. Major comments.  1) Why do the authors report the study as retrospective particularly in the title? The data appears to be collected prospectively including the Global Ratings of Change questionnaire? 2) A major limitation is the lack of a sample size calculation for the GRS categories and multiple time points. It is clear (and the authors acknowledge) that there were too few patients in the 'large' difference categories to be interpreted but it's not clear what other data may have type I or II errors within - there are likely to be many with the multiple comparisons being analysed. This needs to be highlighted early in the limitations section. 3) Why did the authors decide to combine the anchor and distribution methods? These can infer different constructs - the former a 'MCID' if it's linked to either clinical or patient preference data whereas the latter is a statistical analysis related to effect size. Appreciate other authors have also combined but suggest adding the limitations to doing this in the discussion. 4) There is some repetition between data described in the text, table and figure which should be reduced. 5) The authors refer to determining whether a change has occurred within an individual patient as a potential use for the data. The MCID derived from a group shouldn't really be extrapolated to an individual patient - understanding the repeatability of the data will inform whether or not a change has occurred within an individual. Suggest removing comment Pg 6 Line 18-20, and line 278. 6) Other authors have included sensitivity and specificity analysis for 'improvers' (all categorised together) vs 'no change' and then ROC analysis - could the authors consider performing this (also vs 'deteriorators') as it may improve their statistical power? 7) the results in the abstract should include the range for the weighted thresholds for the improvement similar to how the values for deterioration are presented? Minor comments  1) Add why the MCID may be higher for PR vs RCP - patients have had to put effort in and therefore may have higher expectations of an improvement? 2) Suggest adding in the introduction that MCIDs are particularly relevant for questionnaires whereby a score or change in score isn't intuitively meaningful
---

	3) The authors make little comment about the comparative responsiveness of the questionnaires to PR or in clinical settings? Suggest adding something in the discussion (maybe around line 220). 4) line 25-28 suggest adding 'for the CAT score 2.00 - 3.00 ...' or '2.00 - 3.00 ... units' 4) line 25 rephrase to - 'the estimated MCID for the CAT score is 2.00 ...' 5) line 51 rephrase to - 'Details have been presented previously ...' 6) line 59 change send to sent 7) line 122, 127 the data is in the table so suggest remove from text 8) line 126 remove knowledge 'according to the administrative records'. 9) line 141 - it might make it easier to synthesise the data if the questionnaire the data was referring to was mentioned at the beginning of each sentence rather than at the end e.g. 'after PR a score on the CAT questionnaire was ... for improvers for patients who deteriorated ...' 10) line 148 indicate where this data is presented 11) Pg 12 there is alot of repeated data from the table 1 and 2 suggest remove 12) T2, T3, T5 are not intuitive (presumably for the primary study there was a T4) suggest either changing T5 to T4 or change to T3, T6, T12 indicating the intervals in months. 13) Suggest including the data which is not significant in tables. 14) Pg 15 includes alot of data which is also within the tables - suggest reduce repetition. 15) Table 5 - should include all results regardless of significance 16) line 236 '...large change was too low with wide confidence intervals to enable valid conclusions' 17) line 281 suggest leave at 'One cannot directly transform the thresholds for improvement into deterioration'. 18) abstract - suggest change 'follow-up moment' to either just 'follow-up' or 'follow-up visit'.
--	---

VERSION 1 – AUTHOR RESPONSE

Reviewer #1

Comments to the author

1. Line 7-8 In the Abstract, the introduction to the study requires some slight changes to reflect the reason behind the study and introductory sentence was too abrupt.

Thank you very much for your suggestions. We have adjusted the introductory paragraph in the abstract to better reflect the rationale and background of our study. Please note that this will increase the number of words for the abstract from originally 292 words up to 367 words.

2. Line 15-26: Methodology needs to be clearer in the way it reads reflecting a better explanation of how the patients were recruited.

We have adjusted the methods section of the abstract to better explain the recruitment and inclusion of patients in both of our studies. We hope that this makes it better understandable for the reader. Please note that this will increase the number of words for the abstract from originally 292 words up to 367 words.

3. Literature Review: In this section, one which looked into the importance of noting deterioration as well as improvements, there is a lack of stating the importance of such a study to the clinical field.

We have added references and highlighted the need for evaluating MCIDs for deterioration in health status more clear in our introduction section of the manuscript (see lines 17-28, line 33-34, see the new references 20, 32-34). A systematic review by Alma et al. (2018) in the European Respiratory Journal clearly demonstrated that there are currently no studies investigating the MCIDs of the CAT, CCQ and SGRQ regarding deterioration (see new manuscript reference 20). Please note that in the remainder of the manuscript the numbering of the references has been adjusted. All changes are highlighted in the manuscript in red.

4. Lines 37 - 44 There is the need to explain the recruitment of patients better arranging the sentences to read in a better manner.

We have re-arranged the paragraph on the study subjects in our methods section (see lines 39-52). We hope that this makes the recruitment of patients better readable and understandable.

5. In Line 51, 52 and 54, it is stated that spirometry tests were carried out. How was this done when it is stated a bit before that patients were not actively involved.

We have added to the text that the spirometry test for study one (PR) were taken in the clinic. For the Routine Clinical Practice (RCP) patients in study 2 the latest spirometry results were obtained from the including general practitioner or the pulmonary physician after approval of the participant. We have included this in the lines 59 and 61-62.

Reviewer #2

Comments to the author

Thresholds for Clinically Important Deterioration versus Improvement in COPD health status during Pulmonary Rehabilitation and Routine Clinical Practice: A Retrospective Analysis. The authors present data from two separate studies regarding the MCID of commonly used questionnaires to assess health status in patients with COPD over multiple time points up to 1 year after pulmonary rehabilitation and during routine clinical practice.

The paper is overall easy to read, includes a reasonably large data set >650 patients over two different settings and the benefits over existing research are the inclusion of both improvement and deterioration assessed over multiple time points. Both anchor and distribution methods are used.

Major comments.

1. Why do the authors report the study as retrospective particularly in the title? The data appears to be collected prospectively including the Global Ratings of Change questionnaire?

Thank you very much for raising this issue. Reviewer #2 is indeed correct that both studies and their data collection should be classified as prospective research. Indeed the data has been collected to primarily or secondarily determine MCIDs of health status questionnaires in patients with COPD. The reason why we wanted to focus on the retrospective element of the analysis was that the Global Ratings of Change questionnaires required patients to assess their health status changes retrospectively. This could perhaps be better defined as a "retrospective approach". However, to avoid further confusion, we have adjusted the title to "results from prospective research" (see title page) and made clear throughout the manuscript in the methods (see lines 39, 63, 68, 78) and discussion (see line 254) that the evaluation of change was judged retrospectively by the patient.

2. A major limitation is the lack of a sample size calculation for the GRS categories and multiple time points. It is clear (and the authors acknowledge) that there were too few patients in the 'large' difference categories to be interpreted but it's not clear what other data may have type I or II errors within - there are likely to be many with the multiple comparisons being analysed. This needs to be highlighted early in the limitations section.

Yes it is indeed unfortunate that the categories for moderate and large changes are small. The original sample size calculations for both studies were performed prior to the studies and did not include categorization of the GRCs. The sample size calculation for study one was based upon the randomized controlled trial for COPD patients during PR receiving inspiratory muscle training vs. sham training. For study 2, the sample size was determined based upon the level of the expected MCID to be measured as a difference. We have highlighted this limitation more strongly in the strengths and limitations section of our discussion (see lines 264-266). We have also highlighted this in the article summary section concerning the study's limitations too.

3. Why did the authors decide to combine the anchor and distribution methods? These can infer different constructs - the former a 'MCID' if it's linked to either clinical or patient preference data whereas the latter is a statistical analysis related to effect size. Appreciate other authors have also combined but suggest adding the limitations to doing this in the discussion.

This is most definitely an important issue raised by reviewer #2. We have combined both anchor- and distribution based methods into a weighted score, since both techniques are currently applied to estimate MCIDs. It is recommended to combine both types of approaches in measuring an instrument's MCID (see manuscript reference 22). That is why we have combined them. We have however included this consideration in the limitations section of our discussion though (see lines 267-271).

4. There is some repetition between data described in the text, table and figure which should be reduced.

We have removed some of the results written in the text to avoid repetition between text and tables/figures (see also comments #8, #12 and #15 of reviewer #2 – see page 5 of the rebuttal letter).

5. The authors refer to determining whether a change has occurred within an individual patient as a potential use for the data. The MCID derived from a group shouldn't really be extrapolated to an individual patient - understanding the repeatability of the data will inform whether or not a change has occurred withing an individual. Suggest removing comment Pg 6 Line 18-20, and line 278.

This is definitely correct. MCIDs are determined at the group level and this is difficult to extrapolate to the individual patient. However we feel that the evaluation and interpretation of individual change is an important subject to the physician in clinical practice. Although not ideally, the MCID might give an interpretation of the importance of change for the individual (taken into account the confidence intervals for instance). Up to date we have no method to evaluate individual changes in a quantitative way. Former lines 18-20 have been rewritten into “patients” instead of “patient’s” to highlight the group’s perspective of the MCID (see new lines 24-25). Former line 278 has also been altered to describe that MCIDs are valid at the group level (see revised lines 274-275 and line 285).

6. Other authors have included sensitivity and specificity analysis for 'improvers' (all categorised together) vs 'no change' and then ROC analysis - could the authors consider performing this (also vs 'deteriorators') as it may improve their statistical power?

Thank you very much for this suggestion. We have considered adding data from this analysis. However we feel that, in the light of the already very large amount of data, this would make the article too complex to be easily understood by the reader. We (and also other authors) have used both the mean change method (like in the current study) and ROC curves in a similar publication and we did not find major differences between the results from both approaches (see manuscript references 9, 12 and 14). We are therefore confident that the current data would represent an accurate picture of the thresholds for clinically relevant change.

7. The results in the abstract should include the range for the weighted thresholds for the improvement similar to how the values for deterioration are presented?

This suggestion has been adjusted in the results section of the abstract. Please note that this will increase the number of words for the abstract from originally 292 words up to 367 words.

Minor comments

1. Add why the MCID may be higher for PR vs RCP - patients have had to put effort in and therefore may have higher expectations of an improvement?

We have added a statement on the difference found between settings (PR vs. RCP) in the discussion (see lines 210-212).

2. Suggest adding in the introduction that MCIDs are particularly relevant for questionnaires whereby a score or change in score isn't intuitively meaningful

Thank you for your interesting suggestion. We have incorporated this line in the introduction of our manuscript (see lines 12-13).

3. The authors make little comment about the comparative responsiveness of the questionnaires to PR or in clinical settings? Suggest adding something in the discussion (maybe around line 220).

We have included a statement about the comparison of the study settings (PR vs. RCP) in the methods section line 70, 106 and 119. Furthermore we have added in the discussion a statement on the differences found between both settings (see lines 210-212).

4. line 25-28 suggest adding 'for the CAT score 2.00 - 3.00 ...' or '2.00 - 3.00 ... units'

This has been adjusted in the introduction of the manuscript (see lines 29-30).

5. line 25 rephrase to - 'the estimated MCID for the CAT score is 2.00 ...'

This has been adjusted in the introduction of the manuscript (see lines 29-30).

6. line 51 rephrase to - 'Details have been presented previously ...'

This has been adjusted in the methods section of the manuscript (see line 58).

7. line 59 change send to sent

This has been adjusted in the methods section of the manuscript (see line 67).

8. line 122, 127 the data is in the table so suggest remove from text

The baseline data (as shown in Table 1) have been removed from the text in the results section regarding patient characteristics (see lines 126-131).

9. line 126 remove knowledge 'according to the administrative records'.

This sentence has been removed as suggested (see line 128 and 130).

10. line 141 - it might make it easier to synthesise the data if the questionnaire the data was referring to was mentioned at the beginning of each sentence rather than at the end e.g. 'after PR a score on the CAT questionnaire was ... for improvers for patients who deteriorated ...'

Thank you for this suggestion. We have incorporated this revised structure in the results section (see lines 144-149).

11. line 148 indicate where this data is presented

This data is not presented in details in tables/figures elsewhere, since this was not the primary aim of our study. It was purely analyzed upon to investigate whether we could find patterns that explained why certain patients improved or deteriorated on the respective health status questionnaire (see methods lines 104-105 and see results lines 150-153).

12. Pg 12 there is a lot of repeated data from the table 1 and 2 suggest remove

We have removed a large part of the text that presented the results from table 1 and 2 and just only made reference to the tables (see lines 126-131; 140-149). This was also discussed in comments #4 and #8 of reviewer #2.

13. T2, T3, T5 are not intuitive (presumably for the primary study there was a T4) suggest either changing T5 to T4 or change to T3, T6, T12 indicating the intervals in months.

We have adjusted the T2, T3 and T5 into respectively T3, T6 and T12 (indicating the interval in months) to make it more clear for the reader. This has been adjusted in the methods (see line 99) and throughout the results section of the manuscript. It has also been adjusted in the abbreviations list. All changes are highlighted in red.

14. Suggest including the data which is not significant in tables.

This would include especially the non-significant results from the moderate and large change groups in tables 4-6. We have chosen not to incorporate these non-significant results since this include results from the moderate and large change groups where simply a too small N did not lead to significant results. It would not make sense to draw conclusions using non-significant data with such a small N. Furthermore we feel that more data (which is non-significant) would increase the burden of information in tables 4-6 and make it less clear in our opinion.

15. Pg 15 includes a lot of data which is also within the tables - suggest reduce repetition.

We have removed part of the text on page 15-16 (former page 15) to reduce repetition between the text and tables 4-6 (see lines 166-176). We have eliminated especially the text that describes the results from the moderate and large change groups.

16. Table 5 - should include all results regardless of significance

Please note our feedback on comment #14 from reviewer #2 regarding this matter (see page 5 of this rebuttal letter).

17. line 236 '...large change was too low with wide confidence intervals to enable valid conclusions'

Thank you for this suggestion. We have adjusted this sentence in our discussion (see line 227-228).

18. line 281 suggest leave at 'One cannot directly transform the thresholds for improvement into deterioration'.

We have removed the remainder of the sentence (see revised lines 278-279).

19. abstract - suggest change 'follow-up moment' to either just 'follow-up' or 'follow-up visit'.

This suggestion to change “follow-up moment” into “follow-up” has been adjusted in the methods section of the abstract. It has also been adjusted throughout the manuscript and highlighted in red where required.

VERSION 2 – REVIEW

REVIEWER	Anabel Sciriha Malta
REVIEW RETURNED	17-Mar-2019

GENERAL COMMENTS	There has been a good improvement in the level and presentation of this article
---

REVIEWER	Rachael Evans Glenfield Hospital Leicester, UK
REVIEW RETURNED	27-Feb-2019

GENERAL COMMENTS	The authors have thoroughly addressed my comments.
--